# The Role of the Root in Spoken Word Recognition in Hebrew: An Auditory Gating Paradigm

**DOI:** 10.3390/brainsci12060750

**Published:** 2022-06-07

**Authors:** Marina Oganyan, Richard A. Wright

**Affiliations:** Department of Linguistics, University of Washington, Seattle, WA 98195, USA; marina0@uw.edu

**Keywords:** spoken word recognition, morphology, Hebrew

## Abstract

Very few studies have investigated online spoken word recognition in templatic languages. In this study, we investigated both lexical (neighborhood density and frequency) and morphological (role of root morpheme) aspects of spoken word recognition of Hebrew, a templatic language, using the traditional gating paradigm. Additionally, we compared the traditional gating paradigm with a novel, phoneme-based gating paradigm. The phoneme-based approach allows for better control of information available at each gate. We found lexical effects with high-frequency words and low neighborhood density words being recognized at earlier gates. We also found that earlier access to root-morpheme information enabled word recognition at earlier gates. Finally, we showed that both the traditional gating paradigm and gating by phoneme paradigm yielded equivalent results.

## 1. Introduction

In this study, we investigate online morphological processing of spoken Hebrew, a templatic language, using the traditional gating paradigm and a novel phoneme-based adaptation of the paradigm. In particular, we investigate online lexical processing of templatic words and online morphological processing and the role of the root in spoken word recognition.

When investigating the role of morphological complexity in spoken word recognition, it is important to take into account the relative structure of the language’s morphology. The structure of concatenative languages, such English, Japanese, or Swahili (and many other languages), differs from that of templatic languages, such as Hebrew and Arabic (and a handful of other languages), in the distribution of lexical and derivational morphemes. In concatenative languages, complexity is largely expressed through affixation, with morphemes occurring sequentially, while in templatic languages, lexical and derivational morphemes are interleaved (the differences are illustrated in Table 1). In Semitic languages, the combination of the *root*, the semantic/lexical part, and *template*, the derivational part, form the stem for all Semitic words (excluding non-Semitic loanwords). This makes all Semitic words inherently morphologically complex. While affixation is used in Semitic languages, it is restricted to inflectional morphology.

Morphological processing of affixed words in concatenative languages has been widely studied in visual word recognition, resulting in strong evidence for (at least some) decomposition of words into composite morphemes during word recognition, e.g., [1] (English), [2] (French), [3] (Finnish), [4] (Japanese). As with concatenative words, there is evidence that templatic words are parsed into their morphological units (namely the root and template) during visual word recognition [5,6,7,8,9,10]. A smaller but also substantial body of research exists for word auditory word recognition in concatenative languages with similar findings (e.g., [11,12]).

In Hebrew and Arabic, the root morpheme has been shown to be key to word recognition in a way that is different from concatenative languages. For example, in concatenative words, letter transpositions do not inhibit priming, while in templatic words, transposition of root letters inhibits priming ([7] (Hebrew), [8] (Arabic)). Additionally, while in concatenative languages, priming occurs at the stem or affix level, templatic words can be primed by words with shared roots regardless of semantic relationship (e.g., [13]).

While a small amount of research has been conducted on auditory word recognition in Semitic languages, it has primarily used offline paradigms such as priming and auditory masking. For example, Geary and Ussishkin [10] found that root priming in templatic words extends to the auditory domain. Additionally, Oganyan, Wright, and Herschensohn [14] found that noise-masking root morpheme sounds in auditory stimuli makes a word more difficult to recover than noise-masking template sounds. Extending word recognition research to the auditory domain is important because it strengthens our understanding of the role of morphology by reducing the potential for orthographic interference. In this study, we use gating paradigms to investigate real-time auditory processing of words where full root information (the *root completion point* RCP) is presented either earlier or later in the signal to test the relative importance of the root and template in the timing of word recognition.

An important aspect of auditory perception is the linearity of the signal and the ability to amend perception of the word as more of the signal becomes available. For example, when a Hebrew listener hears the onset of a word beginning with a/k/, as in the Hebrew word (/katav/כתב), there are a large number of /k/-initial lexical competitors; however, as the auditory word progresses, the number of competitors narrows. This aspect of auditory perception has given rise to models of spoken word recognition such as Trace [15] and Cohort [16,17] and, more recently, cognitive network approaches (e.g., [18]). To explore this aspect of spoken language, we employ an auditory gating paradigm [19].

The *gating paradigm*, originally developed by Grosjean in 1980 [19], exposes increasing information from the speech signal. This paradigm mirrors the temporal unfolding of speech information in the auditory perception process while permitting the experimenter to probe the time course of auditory perception and word recognition at different time points. One finding in Grosjean’s [19] study that is relevant to the current investigation is that word duration, measured in number of syllables, affects word recognition time, where the greater the syllable count, the later word recognition takes place. Additionally relevant to this investigation is his finding that words with high usage frequency are recognized earlier than their less-frequent counterparts. A later study by Metsala in 1997 [20] used the same gating paradigm to extend the study of lexical effects to include phonological-neighborhood density, the number of phonological competitors, and its interaction with usage frequency. In his study, he used a two-by-two design crossing *frequency* (low and high) with *density* (low and high). He found that for high-frequency sets, low-density words were recognized earlier than high-density words, but for low-frequency sets, high-density words were recognized sooner than low-density words.

The first set of goals of this paper is to replicate and extend findings of lexical effects for usage frequency and phonological-neighborhood density, observed using the gating paradigm in concatenative languages [19,20], to Hebrew and to test the effect of templatic morphology on the process. In particular, we evaluate whether earlier access to complete root information (RCP) relative to the uniqueness point (UP), the point in the word where there are no possible auditory competitors, leads to earlier word identification. We hypothesize that lexical effects will extend to spoken Hebrew in a way that is analogous to studies of English, with high-frequency words being recognized sooner than their low-frequency counterparts and with interactions between neighborhood density and word frequency.

A second goal of this paper is a methodological one: to test a novel alternative version of the gating paradigm with gates set by perceptual phoneme boundaries rather than traditional fixed 20–60 ms windows. One drawback of traditional gating methods with fixed-window durations is that stimuli have to be very carefully matched to avoid consonant-manner effects interfering with observations. The reason for this is that different consonant manners have different acoustic time courses, and therefore, a fixed-window duration will reveal very different amounts of lexical information if the stimuli are not matched. This severely limits the number and variety of stimulus words since they have to have the same manner sequences within comparison groups. On the other hand, a phoneme-gating paradigm allows for the use of words that are not matched in this manner. A second drawback to the traditional fixed-window paradigm is testing time; with short, fixed windows, a word is broken into a large number of gates, and many of the gates are redundant with previous ones in terms of phonemes. If our novel phoneme-gating paradigm is equivalent to the traditional gating paradigm, as we hypothesize, it will greatly increase the number of possible stimuli, reduce testing time, and increase the kinds of research questions which can be addressed. One important limitation to the novel approach is that the phonemic gates have to be very carefully applied by thoroughly trained acoustic phoneticians to avoid revealing information about preceding or following phonemes. The necessary expertise will limit who can conduct research with this method.

## 2. Materials and Methods

### 2.1. Stimuli

All stimuli were Hebrew words read from randomized wordlists by a male native Hebrew speaker. The recordings were made using a Zoom H4n professional recorder with an AKG C520 head-mounted condenser microphone in a sound-treated recording booth at the Phonetics Laboratory on the University of Washington campus.

For each word, we calculated the uniqueness point (UP) and the root completion point (RCP). The UP refers to the point in the acoustic signal where a word has no lexical competitors. The RCP refers to the point in the acoustic signal at which all Semitic root information has been completed.

The wordlists included two sets of spoken-word recognition stimuli: (1) *lexical*, which were used to test the effects of usage-frequency and phonological-neighborhood density, and (2) *morphological*, which were used to test the effects of Hebrew morphology. Acoustic stimulus duration ranged from 547 ms to 999 ms with an average of 736 ms. Within stimuli, phone duration ranged from 17 ms to 394 ms with an average of 127 ms. A full list of stimuli used can be found in the Appendix A. Results data are available upon request from the authors.

#### 2.1.1. Lexical Stimuli

Forty nouns were selected in a 2 × 2 stimulus matrix design for neighborhood density and usage frequency. Usage-frequency was taken from the database by Frost and Plaut [21], which is a database of written word-usage frequency based on newspapers. Neighborhood density (ND) is defined using the method established by Charles-Luce and Luce [22] as the edit distance of one phoneme (addition, subtraction, deletion, or substitution). Neighborhood density was calculated using a modified version of the MILA corpus lexicon [23] with phonological transcriptions. The MILA (“word” in Hebrew) corpus lexicon of Hebrew words contains more than 25,000 lexicon items. Half of the words were high-frequency (>16 per million), and half were low-frequency (<4 per million). Half of the words were high-density (>12 neighbors), and half were low-density (<3 neighbors). This resulted in a total of 10 words for each combination: high-frequency, high neighborhood density (HF-HND); high-frequency, low neighborhood density (HF-LND); low-frequency, low neighborhood density (LF-LND); and low-frequency, high neighborhood density (LF-LND) (see Table 2). Words were all equal in length (5 phones), beginning with a root sound and for each word the UP and RCP coincided.

#### 2.1.2. Morphological Stimuli

Thirty nouns were selected for their relative position of RCP and UP to form three conditions split across the stimuli with ten words in each. Words either had root completion precede uniqueness point (RCP < UP), uniqueness point precede root completion (UP < RCP), or the two occurring at the same point (RCP = UP). Words were balanced initially for manner of initial phoneme, frequency, and density (see Table 3). All words began with a root sound.

### 2.2. Gating Paradigms

In the first gating paradigm, which we refer to as the *traditional paradigm*, words were cut into 50 ms segments increasing in length with each gate and with the final segment being the full length of the word. This is the traditional gating paradigm first developed by Grosjean [19].

In the second paradigm, which we refer to as the *phoneme paradigm*, words were cut at perceptual phoneme boundaries with the first segment containing the first phoneme, the second the first two phonemes, and continuing until the last contained the full word. Gates were assigned by two trained acoustic phoneticians and were tested on a native Hebrew speaker to ensure that there was insufficient coarticulatory information for the following speech sound to be recovered.

There are several reasons for exploring a phoneme-based alternative to the traditional gating paradigm. The first is that phonemes vary in their intrinsic duration, so a fixed-gate duration exposes different amounts of acoustic information when words differ in their segmental makeup. Thus, an initial gate of 50 ms for a word starting with a stop followed by a vowel, for example, קיטור (kituʁ), will expose significantly more information than for one with a fricative followed by a vowel, for example שחק (ʃaχak). This difference is illustrated in Figure 1, which shows a pair of spectrograms with the two gating paradigms marked out (50 ms gates and phoneme gates).

As can be seen in comparing the two spectrograms, the initial 50 ms gate in the stimuli reveal very different amounts of information about the word. For קיטור (kituʁ) (left), the initial 50 ms gate reveals the consonant’s release burst and a portion of the following vowel, whereas for שחק (ʃaχak) (right), the onset of the vowel is not revealed until the fourth 50 ms gate. To avoid this problem, traditional gating paradigms must restrict the manner of articulation of consonants to have comparable information flow across time. Therefore, gating by phonemes greatly increases the number of possible stimuli available to the researcher. Moreover, the number of gates needed per word greatly decreases because the duration of most consonants and vowels is longer than 50 ms. This can be seen in שחק (ʃaχak) (right), where the traditional paradigm with 50 ms gates requires 15 gates, whereas when gating by phoneme, only 5 are needed. This results in a threefold reduction in testing time. Finally, the traditional gating paradigm is difficult to apply for a research question where control of access to phonemic information is important, such as in the morphological research in this study. This is because the arbitrary nature of the gates means that the information at each gate is difficult to control. A particular gate may contain only partial information about a speech sound, while a different gate may contain information about more than one speech sound if it straddles a boundary. Having more control over the type of information presented at each gate, as the phonemic paradigm does, allows for these types of questions to be addressed.

### 2.3. Participants and Procedure

The experiment was run using an online version of Psychopy (version 2020.1.3) [24] running on the Pavlovia platform (https://pavlovia.org/ accessed from 1 June 2020 to 7 Febuary 2022). All participants were recruited using the Prolific platform (https://www.prolific.co/ accessed from 1 June 2020 to 7 Febuary 2022). Using Prolific’s screening, participants were screened for being native speakers of Hebrew who had grown up in a monolingual household. They were also screened for reporting having normal hearing. Participants who reported living outside of Israel for more than two years were excluded from the study. All instructions were in Hebrew.

Participants wore headphones of their choosing, reporting the brand and model as part of an initial survey to ensure headphone use. Headphone use was set as a technical requirement on the Prolific platform for participation in the study. Participants chose a comfortable listening level on their own devices.

The two gating paradigm experiments were run as independent experiments. Within each gating paradigm experiment, all stimuli were divided into five lists with a balanced sampling from each of the different stimulus types (the four conditions for lexical and three for morphological). Each list was posted on Prolific as a separate task within its relevant experiment. Participants were restricted to participation in only one of the two experiments and were able to complete 1 to 5 of the tasks in their chosen experiment.

For the lexical (frequency by density) experiments, a total of 130 participants took part. For the traditional gating paradigm, 57 participants took part (35 male, 22 female). Participant ages ranged from 18 to 42 years with an average age of 26.5 years (5.7 standard deviation). Each word was responded to between 26 and 35 times (avg. 31). In the gating by phoneme paradigm, there were 73 participants (34 male, 39 female). Participant ages ranged from 18 to 59 years with an average age of 29.5 years (7.5 standard deviation). Each word was responded to between 32 and 40 (avg. 36) times.

In the morphological experiments, a total of 128 participants took part. For the traditional gating paradigm, 56 participants took part (34 male, 22 female). Participant ages ranged from 18 to 42 years with an average age of 26.4 years (5.9 standard deviation). Each word was responded to between 25 and 36 (avg. 31) times. For the phoneme gating paradigm, 72 participants took part (34 male, 38 female). Participant ages ranged from 18 to 59 years with an average age of 29 years (7.7 standard deviation). Each word was responded to between 31 and 39 (avg. 36) times.

In both paradigms, words were presented incrementally increasing in duration with each gate. At each gate, participants were asked to guess the identity of the word and give a confidence value for their guess.

### 2.4. Analysis

Each stimulus was analyzed using both the *recognition point* (RP) and the *isolation point* (IP). The term recognition point here is the point when the word was first guessed correctly, while the isolation point is the point at which the participant guessed the word without changing the guess at subsequent gates. Both RP and IP were used to have a more thorough comparison between gating paradigms since different researchers used one or the other of these with the traditional gating paradigm (e.g., [19,25] IP, [20] RP). Because the IP and RP are largely equivalent, the IP results are reported in the body of the text (RP results are reported in separate tables). Any differences are discussed in the results and discussion sections.

#### 2.4.1. Preprocessing

In preprocessing the results, we established inclusion criteria for responses. Participants were excluded if they stated that Hebrew was not their first language or if they had more than one first language. Participants were also removed if a valid participant ID was missing, indicating an improper submission; all such entries contained no valid guesses. Valid guesses were those with at least one Hebrew letter in the guess. If a participant had no valid guesses for any of the gates of a particular word, responses to that word were omitted for that participant. See Table 4 for a summary of all omitted data as raw counts. Four stimulus words were excluded from the lexical experiment: two for not meeting criteria of having identical gating and uniqueness points and two for having close homonyms. Results were processed with a script, which removed (1) all entries by participants not meeting inclusion criteria, (2) erroneous stimuli, and (3) a participant’s responses to any word with no valid guesses. In addition, the script marked as correct all non-ambiguous typos or misspellings. Due to the vowelless nature of the Hebrew spelling system, incorrectly typed words could only be allowed if there was a clear typo (e.g., inclusion of a non-letter key such as a number or shift key) or confusion between two letters with the same sound, which did not form a different word.

#### 2.4.2. Statistical Analysis

Responses to lexical stimuli were analyzed both in terms of absolute IP, as has been traditionally done in the gating paradigm (e.g., [19,26]), and in terms of difference between IP and UP (IP–UP). This difference measure is useful for controlling for variations in acoustic-word duration. That is, while there may be identical numbers of letters in a written word and therefore no durational difference, different speech sounds may exhibit small differences in duration, introducing noise into the estimation of word recognition point. For the lexical stimuli, no statistically reliable difference was expected between the results for the two measures (IP, IP–UP) because stimulus word length was relatively easy to control for. The research question for the lexical stimuli also lends itself to both IP and IP–UP measure analyses. However, for the morphological stimuli, the stimulus design resulted in word length differences (see Table 3). Furthermore, the research question about the relative ordering of the UP and the RCP did not lend itself to analysis in terms of absolute IP. Therefore, for morphological responses, data were analyzed using only the IP–UP difference measure. For the traditional paradigm, the IP–UP difference was measured in ms, while in the gating by phoneme paradigm, it was measured by gates.

The results of the lexical experiments, IP and IP–UP, were submitted to 2 × 2 linear mixed effects (LMER) models with *density* and *frequency* as fixed effects and *participant* as a random intercept (R formula = IP or IP-UP~Freq * Density + (1|Participant). Two additional comparisons were made, one for frequency (high vs. low) and the other for neighborhood density (high vs. low), using linear mixed-effects regression (LMER) models with *type* as the independent variable, IP or IP–UQ as the dependent variable, and participant as a random intercept (R formula = IP or IP-UP~Freq or Density + (1|Participant). To compensate for potential interactions between neighborhood density and frequency, where a frequency effect can mask neighborhood density effects (e.g., [20]), comparisons for neighborhood density were also made within low-frequency and high-frequency sets using an analogous LMER Model.

In the morphological experiments, the IP–UP differences were submitted to an ANOVA with three condition types: RCP < UP, RCP = UP, and UP < IP. An LMER model was additionally used to compare the three conditions (RCP < UP, RCP = UP, and UP < IP). A second LMER model was used to compare only two condition types: RCP < UP and UP < RCP. In both LMER models, the dependent variable was the IP–UP difference, the independent variable was condition type (RC < UP, RC = UP, UP < IP, or RC < UP, UP < IP), and participant was a random intercept.

## 3. Results

### 3.1. Gating by Time: Lexical

#### 3.1.1. Isolation Points

The results of the lexical experiments are summarized in Table 5. Average isolation points by type were HF-HND 413 ms, HF-LND 385 ms, LF-HND 470 ms, and LF-LND 456 ms. The 2 × 2 LMER revealed an effect for *frequency* (t = 5.781, *p* < 0.001) and for *density* (t = −3.164, *p* < 0.01) but not the *frequency* by *density* (t = 1.127, *p* < 0.26) interaction. Additional LMER models were run separately for *frequency* (H vs. L) and for *density* (H vs. L), with participant as a random intercept. An additional LMER model was run for *density* (H vs. L) within high and low frequencies. LF words were identified on average 63 ms slower than HF words (t = 9.253, *p* < 0.001). Overall, LND words were identified on average 17 ms sooner than HND ones (t = 2.38, *p* < 0.05). The effect was carried by differences for low-frequency words, with no significant effect for high-frequency words (t = −1.42, *p* < 0.156). Low-frequency words with LND words were recognized 28 ms sooner relative to UP than HND on average (t = −2.948, *p* < 0.01).

#### 3.1.2. Difference Isolation Point to Uniqueness Point

On average, HF-HND words were identified 92 ms before, HF-LND 96 ms before, LF-HND 13 ms after, and LF-LND 34 ms before the UP. The 2 × 2 LMER revealed an effect for *frequency* (t = 10.355, *p* < 0.001) and for the *frequency* by *density* (t = −3.024, *p* < 0.01) interaction but not for *density* (t = −0.498, *p* = 0.619). Additional LMER models were run separately for *frequency* (H vs. L) and for *density* (H vs. L), with participant as a random intercept. An additional LMER model was run for *density* (H vs. L) within high and low frequencies. There was an effect of frequency, with LF words being identified 80 ms later relative to the UP than HF words (t = 11.47 *p* < 0.001). There was also an overall effect for ND with LND words being identified 17 ms sooner than HND words (t = −2.304, *p* < 0.05). The effect was carried by differences at low frequency words, with no significant effect in high-frequency words (t = −0.413, *p* < 0.68) and a significant effect in low-frequency words with LND words being recognized 47 ms sooner relative to UP than HND words on average (t = −4.917, *p* < 0.001).

### 3.2. Gating by Time: Morphological

#### Difference—Isolation and Uniqueness Points

The results of the morphological experiments are summarized in Table 6. On average, RCP < UP words were identified 46 ms sooner, RCP = UP words were identified 15 ms sooner, and UP < RCP words were identified 79.23 ms later than the uniqueness point. The ANOVA revealed a significant effect for type overall (F = 83.906 value, *p* < 0.001). The first LMER model, with type as an independent variable and participant as a random intercept effect, revealed an effect for type: RCP < UP words were identified on average 31 ms before RCP = UP words (t = −3.067, *p* < 0.01) and UP < RCP 95 ms later than the RCP = UP words (t = 9.363 *p* < 0.001). Results from the second LMER model comparing RCP < UP and UP < RCP words revealed a significant effect with UP < RCP identified 125 ms after RCP < UP relative to the uniqueness point (t = 11.93, *p* < 0.001).

### 3.3. Gating by Phoneme: Lexical

#### 3.3.1. Isolation Points

The results for the lexical experiment using phoneme gating are summarized in Table 5. Average isolation points by type occurred at gate 4.125 for HF-HND words, at gate 3.896 for HF-LND words, at gate 4.536 for LF-HND words, and at gate 4.376 for LF-LND words. The 2 × 2 LMER revealed an effect for *frequency* (t = 5.613, *p* < 0.001) and for *density* (t = −3.327, *p =* 0.001) but not the *frequency* by *density* (t = 0.670, *p* = 0.503) interaction. Additional LMER models were run separately for *frequency* (H vs. L) and for *density* (H vs. L), with participant as a random intercept. An additional LMER model was run for *density* (H vs. L) within high and low frequencies. LF words were identified on average 0.435 gate later than HF words (t = 8.513, *p* < 0.001). Overall LND words were identified on average −0.17158 gate earlier than HND ones (t = −3.286, *p* < 0.01). The effect was carried by differences in low-frequency words, with no significant effect in high-frequency words (t = −0.153, *p* < 0.878) and a significant effect in low-frequency words, with LND words being recognized on average 0.160 gate sooner relative to UP than HND words (t = −2.25, *p* < 0.05).

#### 3.3.2. Difference—Isolation and Uniqueness Points

On average, HF-HND words were identified −0.875 gate before, HF-LND −0.887/gate before, LF-HND 0.464 gate before, and LF-LND −0.624 gate before the UP. The 2 × 2 LMER revealed an effect for *frequency* (t = 5.477, *p* < 0.001) but not for *density* (t = −0.162, *p =* 0.871) or the *frequency* by *density* (t = −1.424, *p* = 0.155) interaction. Additional LMER models were run separately for *frequency* (H vs. L) and for *density* (H vs. L), with participant as a random intercept. An additional LMER model was run for *density* (H vs. L) within high and low frequencies. There was an effect for frequency with LF words being identified 0.330 gate later relative to the UP than HF words (t = 6.325, *p* < 0.001). There was no significant overall effect for ND (t value = −1.129, *p* < 0.259). There was an effect for ND low-frequency words, with LND words being recognized −0.160 gate sooner relative to UP than HND words on average (t value −2.25, *p* < 0.05) but no significant effect for ND for high-frequency words (t value −0.153, *p* < 0.878).

### 3.4. Gating by Phoneme: Morphological

#### Difference—Isolation and Uniqueness Points

The results for the morphological experiment are summarized in Table 6. On average, RCP < UP words were identified 0.887 gate before, RCP = UP words were identified 0.241 gate before, and UP < RCP words were identified 0.496 gate after the uniqueness point. An ANOVA with type as the independent variable and IP gate number as the dependent variable revealed a significant main effect (F value = 183.38, *p* < 0.001). The first LMER model, with type as an independent variable and participant as a random intercept effect, revealed an effect for type: RCP < UP words were identified on average 0.646 gate before RCP = UP relative to UP < RCP (t = −8.94, *p* < 0.001) and UP < RCP 0.737 gate after RCP = UP relative to uniqueness point (t = 10.121, *p* < 0.001). Results from the second LMER model comparing RCP < UP and UP < RCP words revealed a significant effect with UP < RCP identified 1.3828 gate after RCP < UP relative to uniqueness point (t = 18.12, *p* < 0.001).

### 3.5. Recognition Point Results Summary

The recognition point (RP) results are summarized in Table 7 (lexical experiments) and in Table 8 (morphological experiments). In both lexical and morphological results in both traditional and phoneme gating paradigms, the results for RP and RP-UP did not differ in significance, magnitude, or direction of effect from those obtained with IP and IP-UP. There were two exceptions to this equivalence in the RP metric for neighborhood density comparisons (H vs. L). The direction of the effect remained the same as in the IP metric (LND words were identified more quickly than words with HND). In the gating by phoneme paradigm, the effect was significant for both high- and low-frequency words (as opposed to low frequency only), and in the traditional paradigm, it was significant for high- but not low-frequency words (as opposed to low-frequency only).

## 4. Discussion and Conclusions

This study represents the first time an auditory gating paradigm has been applied to spoken Hebrew to test lexical and morphological effects in word recognition. Using the gating paradigm allows us to observe how word information unfolds over time in the spoken signal and how lexical and morphological factors interact with auditory word recognition. Hebrew is an interesting test case because the Semitic templatic morphology has been shown, using other methods, to interact with word recognition in a way that is different from concatenative languages. Furthermore, we introduced and tested the phoneme-gating paradigm, which can greatly expand the number of stimuli and which has the potential to expand the kinds of questions a researcher can address using gating.

### 4.1. Lexical Results

Higher-frequency words were recognized at shorter gating times than lower-frequency words both in terms of IP and the IP-UP measures. This result is in line with previous findings in concatenative languages with the gating paradigm [19,20]. That is, less information is needed for a listener to recognize higher frequency words.

For higher-frequency words, there was no statistically reliable effect for neighborhood density. However, for lower-frequency words, words with lower neighborhood density were recognized at earlier gates than words with higher neighborhood density. These findings differ from those by Metsala’s [20] results for English, where for higher-frequency words, low neighborhood density words were identified more quickly than high-density words, and for lower-frequency words, high-density words were identified more quickly than low-density words. It is always complicated to compare results such as these across languages because of the myriad ways in which any two languages may differ. Neighborhood density effects have been shown to differ between languages in previous studies. For example, while high neighborhood density has a facilitatory effect in Spanish [27], in English, high neighborhood density has an inhibitory effect (e.g., [28]). Furthermore, the inherent morphological complexity of Hebrew words may also contribute to the differing results. Therefore, our results should not be taken as a refutation of previous findings but perhaps as a further example of the complexity of comparing lexical effects across languages.

### 4.2. Morphological Results

Words in which the root completion point preceded the uniqueness point (RCP < UP) were identified with less signal information. In contrast, words in which the uniqueness point preceded the root completion point (UP < RCP) needed more signal information. That is, not having all root-phoneme information made it difficult to identify a word correctly. This result replicates, and extends to the online auditory domain, previous findings that the root is important for word recognition in templatic languages. In particular, during the process of recognition, having access to root information may narrow the scope of guesses not just acoustically but also morphologically, allowing for words to be identified with less information. This is an important extension of previous findings that root information plays a crucial role in word recognition in Hebrew, and it is novel in that the gating paradigm has allowed us to observe the time course of the process in comparing the effect of the UP to the RCP.

### 4.3. Paradigms

In both the lexical and the morphological experiments, the effect significance and effect direction did not differ between the two gating paradigms. This suggests that gating by phoneme is an appropriate methodology, at least for addressing certain types of research questions. Being able to gate by phoneme extends the types of research questions that could be addressed with gating, allowing for more careful control of information available to participants at each gate. Furthermore, this adaptation of the paradigm addresses the problem of having to control for the acoustic duration of phonemes across stimuli. We feel that this is a new and powerful research tool. While there are advantages to the phonemic gating paradigm, it much more difficult to apply than the traditional paradigm. Cutting stimuli such that there is access to only one additional phoneme at each gate requires precision and extensive acoustic phonetic training.

### 4.4. Differences between RP and IP Results

RP and IP results differed only in one aspect: the magnitude and statistical significance of neighborhood density effects in low vs. high frequencies with RP/IP as the dependent measures. While neighborhood density effects were only significant at low frequency with IP, with RP, they were only significant at high frequency or in both low and high frequency. Given that the difference between the RP and IP is whether a participant subsequently changed the answer from a correct guess to an incorrect one and then back again, differences with regards to neighborhood density (i.e., potential competitors) are not surprising. These differences may in fact be attributed to the frequency or more likely the location in the signal at which potential competitors (neighbors) appeared for the high- vs. low-frequency stimuli. That is, this difference may be the result of differences between high- and low-frequency words in the position in a word where changing a phoneme created a neighbor. If this position was later in the word for low-frequency words, this may cause more incorrect back-tracking after a correct guess. Thus, measuring from RP-UP, incorporating the uniqueness point at which no neighbors exist instead of just RP eliminates any differences in effects. In related work with the gating paradigm in English, Vitevich [26] found that neighborhood density effects were the result of neighborhood spread (the number of positions in the word at which potential neighbors could occur). The current stimuli were not designed to fully test this prediction, so it is left to future work to address this more rigorously.

### 4.5. Future Directions

In the morphological experiments of this paper, we focused on the role of the root overall and its importance in spoken word recognition. However, the role of roots and templates in recognition of words in Hebrew and other templatic languages is tied not only to the morphemes themselves but also to their productivity. For example, in Hebrew, Farhy, Verissimo, and Clahsen [29] found that morphological root priming occurred in words with a productive verbal template but not with a different, non-productive, verbal template. In Arabic, Boudelaa and Marslen-Wilson [30] found that morphological priming effects were only found in words with productive roots. Thus, taking into account factors, such as the predictability of a morpheme based on its context, could be applied to an investigation of the role of morphology in templatic spoken word recognition. In future work, we plan to extend our research to investigate spoken word recognition based on productivity of these root and template morphemes and their co-occurrence as well as context effects.

## Figures and Tables

**Figure 1 brainsci-12-00750-f001:**
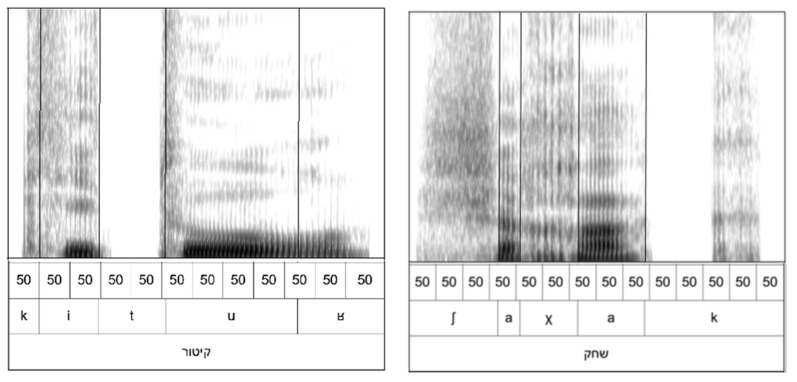
Spectrograms of stimuli קיטור (kituʁ) (**left**) and שחק (ʃaχak) (**right**) illustrating the traditional 50 ms gates and the phoneme gates.

**Table 1 brainsci-12-00750-t001:** Examples of Concatenative vs. Templatic derivational morphology.

	Concatenative (Swahili)				
	**Prefix**	**Root**	**Suffix**	**Word**	**Meaning**
verb	*ku-* (*inf.)*	-pend-	*-a (verb)*	kupenda	to love
noun	*u- (nom.)*	-pend-	-o *(nom*)	upendo	love (n.)
	**Templatic (Hebrew)**				
		**Root**	**Template**	**Word**	**Meaning**
verb		/x/-/k/-/ʁ/	_a_a_ (verbal)	/xakaʁ/	investigated (v. m. past)
noun		/x/-/k/-/ʁ/	mi_ _a_ (nominal)	/mixkaʁ/	research (n.)

**Table 2 brainsci-12-00750-t002:** Lexical Stimuli Properties.

	Phones	Freq	ND	Initial Sound Manner
HF-HND	5	34 (16–64)	14.7 (12–16)	Fricative (6), Stop (4)
HF-LND	5	30.2 (16–52)	1.8 (0–3)	Fricative (3), Nasal (3), Stop (4)
LF-LND	5	1.9 (1–4)	15.1 (12–23)	Fricative (6), Nasal (1), Stop (3)
LF-HND	5	1.7 (1–3)	1.8 (1–2)	Fricative (3), Nasal (4), Stop (3)

**Table 3 brainsci-12-00750-t003:** Morphological Stimuli Properties.

	Phones	Freq	ND	Initial Sound Manner
RCP < UP	5.9 (5–8)	2.6 (1–7)	3.3 (0–8)	Fricative (4), Stop (4), Liquid (2)
RCP = UP	5.7 (5–7)	2.8 (1–7)	3.3 (0–10)	Fricative (4), Stop (4), Liquid (2)
UP < RCP	5.7 (5–7)	2.9 (1–7)	3.3 (0–7)	Fricative (4), Stop (4), Liquid (2)

**Table 4 brainsci-12-00750-t004:** Omitted Data, reported as raw counts.

	By Time		By Phoneme	
	Lex.	Morph.	Lex.	Morph.
Participants —Invalid ID	8	8	7	7
Participants —Language	10	10	10	10
Entries —No Valid Guesses	35	33	29	30
Stimuli	LF HND-רתך,שכירה LF LND-נחיל HF LND-בדיחה	n/a	LF HND-רתך,שכירה LF LND-נחיל HF LND-בדיחה	

**Table 5 brainsci-12-00750-t005:** Lexical Results Summary Table (* indicates significance at *p* < 0.05).

Gating by Time						
	**2 × 2 Freq**	**2 × 2 ND**	**2 × 2 Freq:ND**	**Freq**	**ND in HFreq**	N**D in LFreq**
IP	*	*	NS	H < L *	NS	L < H *
IP-UP	*	NS	NS	H < L *	NS	L < H *
**Gating by Phoneme**						
	**2 × 2 Freq**	**2 × 2 ND**	**2 × 2 Freq:ND**	**Freq**	**ND in HFreq**	**ND in LFreq**
IP	*	*	NS	H < L *	NS	L < H *
IP-UP	*	NS	*	H < L *	NS	L < H *

**Table 6 brainsci-12-00750-t006:** Morphological Results Summary (* indicates significance at *p* < 0.05).

	Overall	Diff from RCP = UP	RCP < UP vs. UP < RC
Gating by Time	*	* RCP < UP faster	* RCP < UP faster
Gating by Phoneme	*	* UP < RCP slower	* UP < RCP faster

**Table 7 brainsci-12-00750-t007:** Recognition point (RP) summaries for the lexical experiments. Results differing from those with IP and IP-UP, as dependent variables are bolded. The asterisk indicates statistical significance.

Gating by Time					
	**HF-HND**	**HF-LND**	**LF-HND**	**LF-LND**	
RP	398 ms	368 ms	454 ms	439 ms	
RP-UP	−107 ms	−113 ms	−3 ms	−52 ms	
Statistics					
	**LMER 2 × 2**	**Freq**	**ND**	**ND in HFreq**	**ND in LFreq**
RP	* Freqt = 5.642, *p* < 0.001 * ND t = −3.318, *p* < 0.001 Freq:ND t = 1.149, *p* < 0.25	* H 62 ms < L t = 9.055, *p* < 0.001	*** L 18 ms < H** **t = −2.577, *p* < 0.05**	*** L 30 ms < H** **t = −3.155, *p* < 0.01**	**L 15 ms < H** **t = −1.473, *p* < 0.141**
RP-UP	* Freq t = 10.035, *p* < 0.001 ND t = −0.667, *p* < 0.505 *** Freq:ND** **t = −2.918, *p* < 0.01**	* H 80 ms < L t = 11.12, *p* < 0.001	* L 18 ms < H t = −2.471, *p* < 0.01	t = −581, *p* < 0.6	* L 48 ms < H t = −4.87, *p* < 0.001
**Gating By Phoneme**					
	HF-HND	HF-LND	LF-HND	LF-LND	
RP	4.074	3.852	4.49	4.302	
ID-UP	−0.926/gate	−0.931/gate	−0.504/gate	−0.698/gate	
	Statistics				
	LMER 2 × 2	Freq	ND	ND in HFreq	ND in LFreq
RP	* Freq t = 5.658, *p* < 0.001 * ND t = −3.179, *p* < 0.01 Freq:ND t = 0.270, *p* = 0.787	* H 0.424 < L t = 8.153, *p* < 0.001	*** L 0.184 < H** **t = −3.474, *p* < 0.001**	*** L 0.222 < H** **t = 3.091, *p* < 0.01**	*** L 0.194 < H** **t = −2.625, *p* < 0.01**
RP-UP	* Freq t = 5.509, *p* < 0.001 ND t = −0.070, *p* = 0.95 **Freq:ND** **t = −1.775, *p* = 0.076**	* H 0.319 < L t = 5.94, *p* < 0.001	t = −0.072, *p* < 0.2	t = −0.005, *p* < 0.95	* L 0.194 < H t = −2.625, *p* < 0.01

**Table 8 brainsci-12-00750-t008:** Recognition point (RP) summaries for the morphological experiments. Results differing from those with IP and IP-UP, as dependent variables are bolded. The asterisk indicates statistical significance.

Gating by Time			
	**RCP < UP**	**RCP = UP**	**UP < RCP**
	−57 ms	−35 ms	67 ms
**Statistics**			
**Overall**		**RC < UP**	**UP < RC**
* F = 94.29, *p* < 0.001	**RC = UP**	* RC < UP 23 ms before t = −2.36, *p* < 0.05	* UP < RC 102 ms after t = 10.49, *p* < 0.001
	**UP < RC**	* RC < UP 124 ms before t = 12.149, *p* < 0.001	
**Gating By Phoneme**			
**Summary**			
	**RCP < UP**	**RCP = UP**	**UP < RCP**
	0.931	0.267	−0.469
**Statistics**			
**Overall**		**RC < UP**	**UP < RC**
* F = 178.06, *p* < 0.001	**RC = UP**	* RC < UP 0.663/gate before t= −8.939, *p* < 0.001	* UP < RC 0.736/gate after t = 9.866, *p* < 0.001
	**UP < RC**	* RC < UP 1.40/gate before t = 18.03, *p* < 0.001	

## Data Availability

Data are available from the authors upon request.

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
