# Peer review of "The Role of the Root in Spoken Word Recognition in Hebrew: An Auditory Gating Paradigm"

_brainsci, 2022, doi:10.3390/brainsci12060750_

Round 1

Reviewer 1 Report

This paper presents the results of a gating experiment in Hebrew.  There are two goals -- the first is to examine the effects of the root and template structure of Hebrew words on word identification, and the second is to compare between two gating paradigms -- by time and by phoneme.

I have three major issues with this paper:

  1. There are no physiological or perceptual manipulations that can index brain functions, so it is not clear to me why it was submitted to a journal with 'brain' in the title.  This is a cognitive psycholinguistic experiment, and there are many journals that would be more appropriate.
  2. The justification for two separate stimuli sets is not good enough -- the difference between the sets is that in the morphological set, the first sound belongs to the root. It is possible to find words in Hebrew which differ in the salience of the root and template structure -- for example -- madbeka vs. mishpacha  - these have the same nominal pattern but the former has an easily identifiable root while the latter does not. This type of comparison would be more relevant to examining the effects or structure. The two sets differ in the salience of the root and template structure -- with the morphological set having a more salient structure -- where the root and the nominal word form are more easily unpacked.  The theoretical justification is incomplete -- do the authors think that more easily decomposed words are easier to perceive? this was not explicitly stated or actually tested in the analyses. 
  3. I didn't understand why the data were analyzed the way they were. If you are using lmer, why not use the frequency and neighborhood density as fixed factors, and also build a model which includes the interaction between them?  Then you can compare the models, and if the one with the interaction is a better predictor than the one with main effects, you know the interaction is there -- and because the design is a 2X2 (high/low freq X high/low density) -- the anova will show you where it is -- Also -- the 4 cell means should be shown in a histogram with standard errors.

The results showed that Hebrew evinces the same patterns as other languages -- with frequency and neighborhood density affecting word identification.  Its not clear whether the words which began with a root letter were identified faster than those which did not. I didn't find the descriptive statistics for the morphological stimulus set -- if this is one of the major questions -- the data need to be there.

The results also showed that there were no differences between the gatimg paradigms.  I was not convinced that the phonemic gates are good enough -- as we know that coarticulation affects the sounds of both consonants and vowels -- so the division into comparative gates is more complex and maybe even impossible -- whereas the division into temporal gates is common to all the words. 

Author Response

Please find attached the responses.

Reviewer 2 Report

## Summary

This paper reports a set of experiments where the gating paradigm was used to study the auditory processing of Hebrew words. Hebrew is templatic language where morphemes are interleaved in the word, offering a unique opportunity to study the relative contribution of the template and the root to the spoken word recognition of such languages. 

The experiments are well designed with careful selection of stimuli. The results are important for furthering the field's understanding of spoken word recognition of templatic languages. In addition, the paper offers a methodological improvement over the traditional gating paradigm by setting the gates at phoneme boundaries. However, it appears that large sections of the results are missing from the manuscript and it is difficult for the current version for be accepted for publication in its current state. It would be helpful also for the authors clarify the following comments and questions I had while reading the manuscript. 

## Additional Comments

My comments are organized in chronological order. 

Page 3

"differing from target by a single phoneme" - Please be more specific, what does “differ” really mean? Do you mean edit distance of 1?

What does MILA stand for? Please provide more details about the usage frequency database used.

How was the threshold for differentiating between high and low frequency, high and low ND determined? What do the values 16 and 4 mean for frequency? 

Page 5

"Participants wore headphones of their choosing" - Was there any attempt to verify that the participant was actually wearing headphones? In future work the authors may consider using Wood et al. (2017)'s headphone test. 

The description of the study design was a bit confusing. Does it imply that some participants provided responses to most of the items while others contributed fewer responses? Did different groups of participants complete either stimuli set/gating type? Do the participant numbers reflect unique subjects across the study or the unique number of subjects who contributed data to that particular condition? It would be good for the authors to provide a bit more clarity on how the studies were implemented.

Page 6

What do the numbers in Table 4 represent? Percentage or raw counts (of entries or participants)?

It would be good for the authors to explain clearly their analytic approach. It is unclear why an ANOVA was run on "Type" when it is evident that the four conditions are crossed (frequency x ND). It would be more straightforward to include the interaction term directly into the linear mixed effects model with effects coding. It is also important to provide more specific information about the LMER models - nature of the random effects structure, the predictors and how they are coded, the outcome variable, these should be very clearly stated in the manuscript. 

Page 7

"random effect" - Is it a random intercept effect or random slope effect?

What do the stars mean in Table 5?

It is also appears that a large section of the Results section is actually missing. I am not sure if this is an oversight when preparing the manuscript, but I had expected a section like 3.2 Gating by phonemes: Lexical, Morphological to follow. Please check. 

typo in "for lower In lower"

Page 8

Given that the differences between RP and IP results were discussed in some detail here it makes sense for these results to be displayed and shown in the main text, rather than being placed in the Appendix.

typo in "Vivetevich"

Page 9

"informational entropy approach" - This idea was introduced with no context or explanation of the approach; please clarify. 

## References 

Woods, K. J., Siegel, M. H., Traer, J., & McDermott, J. H. (2017). Headphone screening to facilitate web-based auditory experiments. Attention, Perception, & Psychophysics, 79(7), 2064-2072.

Author Response

Please find attached the responses.
